# Direct addition of poly-lysine or poly-ethylenimine to the medium: A simple alternative to plate pre-coating

**Alexander Faussner**⬤*, **Matthias M. Deininger**⬤¤, **Christian Weber, Sabine Steffens**

Institute for Cardiovascular Prevention (IPEK), Ludwig-Maximilian-University (LMU), Munich, Germany

¤ Current address: Clinic for anaesthesiology and surgical intensive medicine, RWTH Aachen University, Aachen, Germany
* alexander.faussner@med.uni-muenchen.de

**Data Availability Statement:** All relevant data are within the manuscript and its Supporting information files.

## Abstract

For most cell culture experiments, it is indispensable that the cells are firmly anchored to culture plates, withstanding rinsing steps that can create shear forces and tolerating temperature changes without detaching. For semi-adherent cells such as the common HEK 293 or PC-12 cells, this could so far be obtained by time-consuming plate pre-coating with cationic polymer solutions. We report here, that i) pre-coating with the cheaper poly-ethylenimine (PEI) works as well as the commonly used poly-D-lysine (PDL), but more importantly and novel ii) that simple direct addition of either PEI (1.5 µg/ml) or PDL (2 µg/ml) to the cell culture medium results in strongly anchored HEK 293 cells, indistinguishable from ones seeded on pre-coated plates. Therefore, the replacement of plate pre-coating by direct addition of either PEI or PDL gives comparable excellent results, but is highly labour-, time-, and cost-efficient. Moreover, we could show that addition of PDL or PEI also works similarly well in animal-free culture using human platelet lysate instead of fetal bovine serum. Interestingly, additional experiments showed that strong cell attachment requires only cationic polymers but not fetal bovine serum or human platelet lysate added to the medium.

## Introduction

45 years ago, Mazia et al. reported a protocol for the use of a cationic polymer, poly-L-lysine (PLL), as a pre-coating reagent to assure better attachment of cells to their support—glass at that time—during imaging experiments [1]. Since then, not much has changed. Depending on their type, origin, differentiation status, or growth state, cells can grow in suspension, semi-adherent, or strongly anchored to plastic cell culture plates as cell (mono)layers. For many types of cell experiments, it is of great advantage—if not an absolute requirement—that cells grow as adherent monolayers that can be extensively rinsed and treated with reagents without substantial loss of cells. While many cell types, in particular primary cells, grow by nature strongly attached and thus tolerate even prolonged incubation periods at temperatures ranging from 4°C to 42°C, most semi-adherent cells start floating under these conditions or be lost during the rinsing and incubation steps (personal observations). Among these are the HEK

**Funding:** This work was supported by a grant from Foerderprogramm fuer Forschung und Lehre der LMU Munich (FöFoLe; to A.F. and M. D.) https://www.med.uni-muenchen.de/forschung/foerderprogramme/foefole/index.html The funders had no role in study design, data collection and analysis, decision to publish, or preparation of the manuscript.

**Competing interests:** The authors have declared that no competing interests exist.

(human embryonic kidney) 293 cells that otherwise are an excellent and extensively used tool to investigate the role and function of a wide range of proteins. HEK 293 cells can be easily and efficiently transfected, transiently as well as stably. Most signalling pathways can be studied in HEK 293 cells as they express the protein components of almost all signalling pathways, and they have a low background of e.g. most G protein-coupled receptors (GPCRs). The latter property makes them a great tool to study GPCRs and mutants thereof by heterologous (over) expression. However, HEK 293 cells are growing semi-adherent and show even weakened attachment at lower temperatures. Their anchoring to cell culture support can be potently improved when these are pre-coated e.g. with poly-L/D-lysine (PLL/PDL) as shown the first time by Mazia et al [1]. With this pre-coating, HEK 293 cells tolerate very well several washing steps without detachment as well as prolonged activation of $G\alpha_q$-coupled (over)expressed GPCRs, that commonly results in contraction and weakened attachment of these cells [2, 3].

Working with HEK 293 cells and different PDL pre-coating protocols at hand, we investigated which experimental protocol steps were essential to obtain strong cell adhesion. In this context, we tested initially among other parameters how thoroughly the PDL solution had to be removed after pre-coating to avoid damage to the cells seeded afterwards as some published protocols even recommended washing and drying of the plates [1, 4, 5]. Interestingly, in preliminary experiments we noticed that small amounts of PDL added to the medium to mimic poor removal of the cationic polymer, did not only not harm the cells but rather strongly promoted their attachment.

Replacing the tedious pre-coating protocol by a new technique, the uncomplicated addition of coating solution to the medium while seeding the cells, would have been a huge improvement for cell culture with semi-adherent cells. Therefore, the aim of our study was to investigate this phenomenon in more detail for two commonly used cationic polymers, for PDL and for the cheaper poly-ethylenimine (PEI). The latter is often used for cell transfections [6] or for pre-treatment of glass fibre filters in radioactive binding experiments [7], but rarely for improving cell adherence. Our results show that direct addition of PDL or PEI to the cell culture medium makes the labour- and time-consuming pre-coating of cell culture plates obsolete or can help avoid the expensive acquisition of ready-made pre-coated cell culture dishes. Moreover, this approach works as well for media containing fetal bovine serum as well as for animal free media containing human platelet lysate.

## Materials and methods

### Reagents and constructs

The coding sequences of the human GPR55 and the bradykinin $B_2$ receptor ($B_2R$), both obtained within plasmids from Missouri S&T cDNA Resource Center, were inserted without their respective stop codons between the BamHI and the XhoI restriction sites of the pcDNA5/FRT/TO vector (Invitrogen) using standard cloning techniques. Both receptor sequences were preceded at their N-terminus by a hemagglutinin tag (MGYPYDVPDYAGS) with the last two amino acids (GS) being due to the BamHI restriction site. The $B_2R$ carried in addition a R128A mutation in the highly conserved DRY-sequence of G protein-coupled receptors [8] and could therefore not elicit a $Ca^{2+}$-signal via activation of G protein $G\alpha_{q/11}$. The coding sequence of Renilla luciferase II [9] was inserted between the XhoI and the ApaI restriction sites in order to obtain the fusion constructs GPR55-RlucII and $B_2R$ R128A-RlucII.

### Stock solutions of cationic polymers

For stock preparation 100 mg poly-D-lysine (PDL; Sigma P1024, >MW 300 kDa) was dissolved in 1 l PBS, and 50% (w/v) poly-ethylenimine (PEI; Sigma P3143, MW 750 kDa) was

diluted with water to obtain a 3 mg/ml stock solution. Both stock solutions were filtered through a sterile 0.22 μm filter and stored at 4°C for up to a year.

## Cell culture

The two receptor-RlucII constructs were stably and tetracycline-inducible expressed in Flp-In TREx-293 (hereinafter termed HEK 293) cells using the Flp-In system (Invitrogen) as described previously [10]. The HEK 293 cells were cultured in DMEM high glucose supplemented with 10% fetal bovine serum and 1% penicillin/streptomycin (complete medium; all from Sigma). Stable expression of the constructs in the HEK 293 cells was maintained by selection with hygromycin B (250 μg/ml final; Invivogen). For experiments, the HEK 293 cell monolayers were washed in general with PBS without $Ca^{2+}/Mg^{2+}$ (Sigma), detached with a small amount of trypsin (25 μl/cm$^2$; Sigma). Depending on the experiment, the cells were taken up in a suitable volume of complete medium or medium without any additions (wherever required) and the cell number determined with a Scepter (Merck) using 60 μm tips. After adjusting the cell number to the experimental need by adding the respective medium, the cell suspension was given in wells of a 24-well plate. After addition of the indicated additives (e.g. PDL, PEI, FBS), 200 μl of the respective cell suspensions were transferred with a 12-channel pipette (two tips per well) to a 96-well cell culture plate (black; flat, clear bottom; PerkinElmer) where the wells had been pre-treated or not as specified. After 24 h the expression of the receptor-RlucII constructs was induced with tetracycline (0.5 μg/ml final, added in 25 μl medium). Experiments were in general performed 2–3 days after seeding of the cells, if not indicated otherwise. For experiments with bioluminescence measurements already the next day, the tetracycline was added immediately to the cell suspension before seeding.

For serial cells dilutions, the cell suspension was adjusted to approximately 150.000 cells/ml in the first well of a 24-well plate and then diluted with the medium 1:2 each time for the five wells next to it. Thus, after transfer to the black 96-well plate the highest concentration seeded was around 30.000 cells/well that after 2–3 days of growth would result in a confluent monolayer.

## Adherence test

To estimate the strength of cell adherence, the 96-well tray was put on ice, the medium removed, and the cells were washed within 15 min six times with ice-cold PBS (Biowest,10x solution, pH 7.35) using a 12-channel microplate-washer (CAPP). Care was taken to not direct the washing fluid directly on the cells but against the walls. Thereafter, the plate was kept at room temperature and 40 μl of Hanks-buffered salt solution (HBSS; Sigma) containing 20 mM HEPES, pH 7.2 (Sigma), was added. After application of a white sticker (PerkinElmer) to the bottom of the black 96-well tray and addition of 10 μl of a 5x coelenterazine H solution (Biosynth, 3 μM final concentration in the assay) to the wells, total luminescence was recorded in a plate reader (Tecan infinite F200 Pro) until a peak was reached within a few minutes for each well. The respective peak values were taken as maximal luminescence of each well. For control cells without rinsing, the cell culture medium was removed carefully with a 12-channel pipette before the tray was put on ice, and 40 μl of HBSS were added immediately. This approach helped to avoid almost any loss of cells for the ensuing luminescence measurement. As the integration time of each measurement was one second, at most 48 wells were started at the same time by addition of coelenterazine H in order to keep the deviation of the obtained peak values from the real peak values small.

### Phase-contrast microscopy

Phase contrast images were taken with an inverted Leica DM IL LED microscope (10x or 20x objective) and with a Leica DFC3000 G monochrome digital camera before the medium was removed as otherwise depending on the conditions the cells might have started to detach.

## Results

### Design of experimental set up

To determine the strength of cell attachment to the plastic surface of cell culture plates we used HEK 293 cells that grow semi-adherent and are prone to detach at lower temperatures, which are often required e.g. for radioactive ligand binding experiments [2]. For quantification of the relative amounts of cells on the plates we used existing HEK 293 cell lines that stably expressed constructs of Renilla luciferase II (RlucII) fused C-terminally to either human GPR55 or to a non-G protein-activating mutant of the human Bradykinin $B_2$ receptor, $B_2R$ R128A [8]. The advantage of cells expressing the $B_2R$ mutant is that they do not respond to bradykinin with $Ca^{2+}$ release, contraction and thus *per se* impaired attachment. Trypsin applied to detach the cells can release bradykinin from kininogens present in the serum added to the cell culture medium [11] and thus normally has to be removed by a centrifugation step. However, using either the GPR55- or the $B_2R$ R128A-RlucII fusion constructs, this centrifugation step could be avoided as both are not affected directly or indirectly by residual trypsin activity, even when highly (over)expressed. Making use of these GPR55-RlucII or $B_2R$ R128A-RlucII expressing HEK 293 cells, the relative amounts of cells could be easily quantified via their strong biolumi-nescence signal. This method also allowed detection of low amounts of cells where other tech-niques such as neutral red staining of cells [12] are not sensitive enough. As Fig 1 illustrates, the measured bioluminescence signal correlated well with the dilutions of the cells seeded two or three days before, independent of a pre-treatment of the wells with poly-D-lysine (PDL), although there was a trend to higher bioluminescence values without PDL pre-treatment. The bioluminescence measured in these experiments could reach very high values. Therefore, in order to reduce the interference between wells, we used for the experiments black plates with clear bottom that for the measurement were covered with white stickers. That way we lost approximately 75% of the bioluminescence measured as compared to white plates (S1 Table), but saw an even greater reduction in the interference between the wells: less than 1% of the luminescence was measured in an empty well next to one with luminescence activity for black plates as compared to approximately 4% for white plates (S1 Table).

### Concentration and time dependence of pre-coating with cationic polymers PDL and PEI

To estimate the strength of HEK 293 cell anchoring, adhesion challenging conditions were applied by placing the cell culture plates on ice and rinsing the monolayers six times over a period of 15 minutes with ice-cold PBS using a 12-channel microplate-washer. This challenge resulted in a more or less complete loss of cells when grown on regular cell culture plates (Fig 2A, 0 µg/ml PEI). As controls without rinsing we used cells on the same tray where only the medium had been removed cautiously with a 12-channel pipette followed by addition of the 40 µl of HBSS needed for the bioluminescence assay afterwards. With this procedure, there was almost no cell loss independent of any kind of treatment (Fig 2A–2C).

As mentioned before, a standard method to increase the attachment of semi-adherent cells is the pre-coating of cell culture plates with the cationic polymer poly-L/D-lysine (PLL/PDL) [1]. In addition, also the use of poly-ethylenimine (PEI) had been reported for this purpose

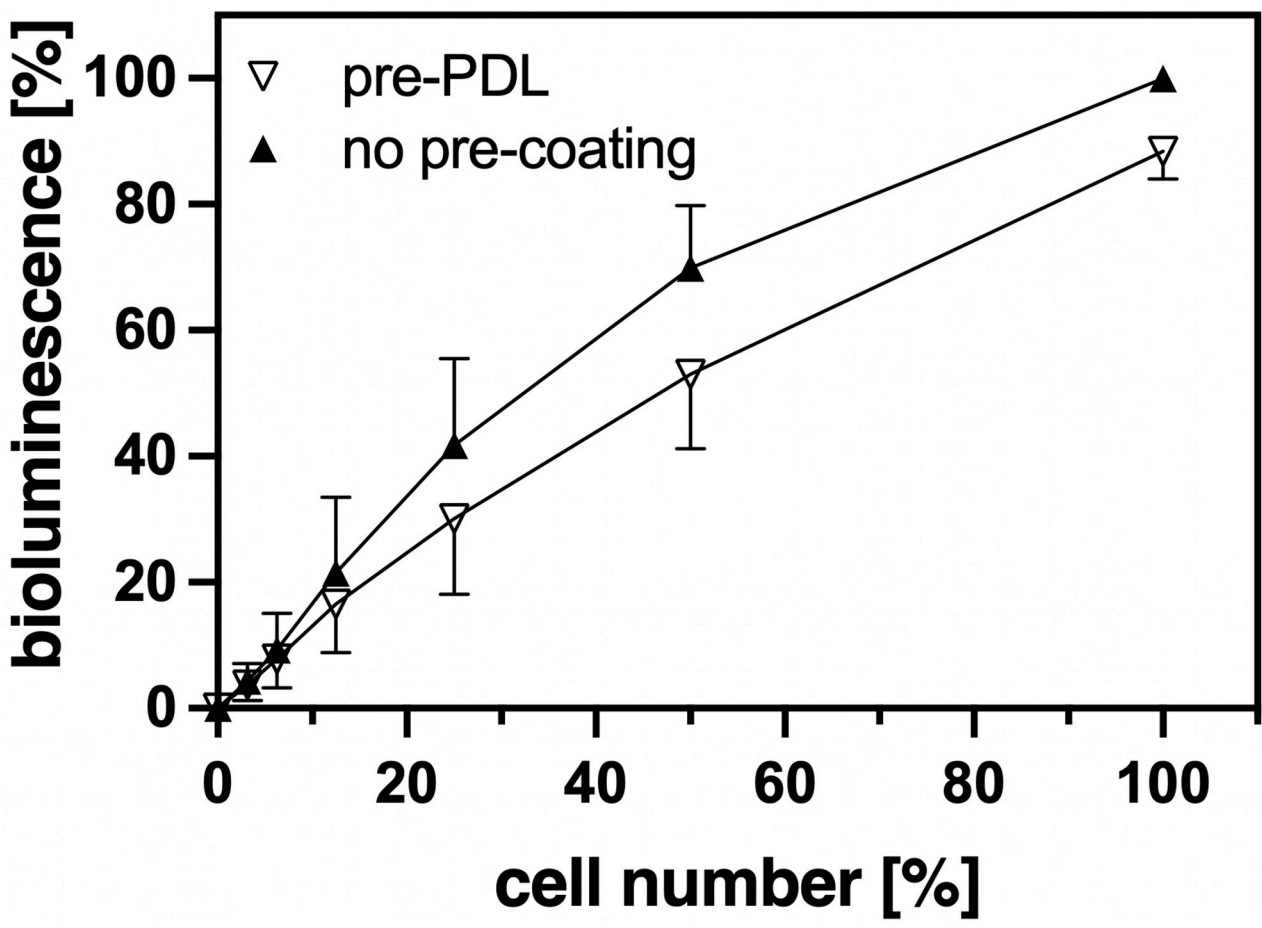

**Fig 1. Correlation between cell density and measured bioluminescence with and without PDL-pre-coating.** Cells expressing either GPR55- or $B_2R$ R128A-RlucII fusion proteins were diluted serially two-fold starting with approx. 30.000 cells per well. The wells were either pre-coated or not with PDL (100 µg/ml, 1 h incubation). After 2–3 d, when the wells with the highest cell number had reached confluency, bioluminescence was determined as described in "Materials and methods". The confluent wells without pre-coating were taken as 100%. Results are presented as Mean ± SD (number n of independent experiments = 5, performed in duplicates).

[12]. Therefore, to reassess existing protocols to promote cell adherence, we focused on these two polymers. Both are easily available and fairly economical, in particular PEI. To begin with, we investigated which concentrations and pre-incubation times sufficed or worked best to guarantee strong adherence of HEK 293 cells and did not require additional rinsing and/or drying of the culture plates as reported before [1, 4, 5]. As not much was known yet about PEI as a pre-coating agent, although it is quite common as a transfection reagent, we compared the effect of increasing pre-coating concentrations of PEI on cell adherence to the one of the standard pre-coating solution of 100 µg/ml PDL [10] (Fig 2A). A PEI concentration of 10–30 µg/ml gave good results comparable to the standard PDL solution, with bioluminescence values reflecting cell adherence of still 80% and more of control after the rinsing challenge (Fig 2A). Higher pre-coating concentrations of PEI required additional rinsing of the wells. Otherwise cell growth and appearance could be strongly affected and the amount of bioluminescence measured would vary considerably. A pre-incubation time of 5 min was found to be enough for both 100 µg/ml PDL (Fig 2B) and 10 µg/ml PEI (Fig 2C) to obtain cell adherence levels that could not be improved by longer incubation times. Thus, as a standard pre-coating procedure, pre-incubation of the plates with either 100 µg/ml PDL or 10 µg/ml PEI for at least 5 min at

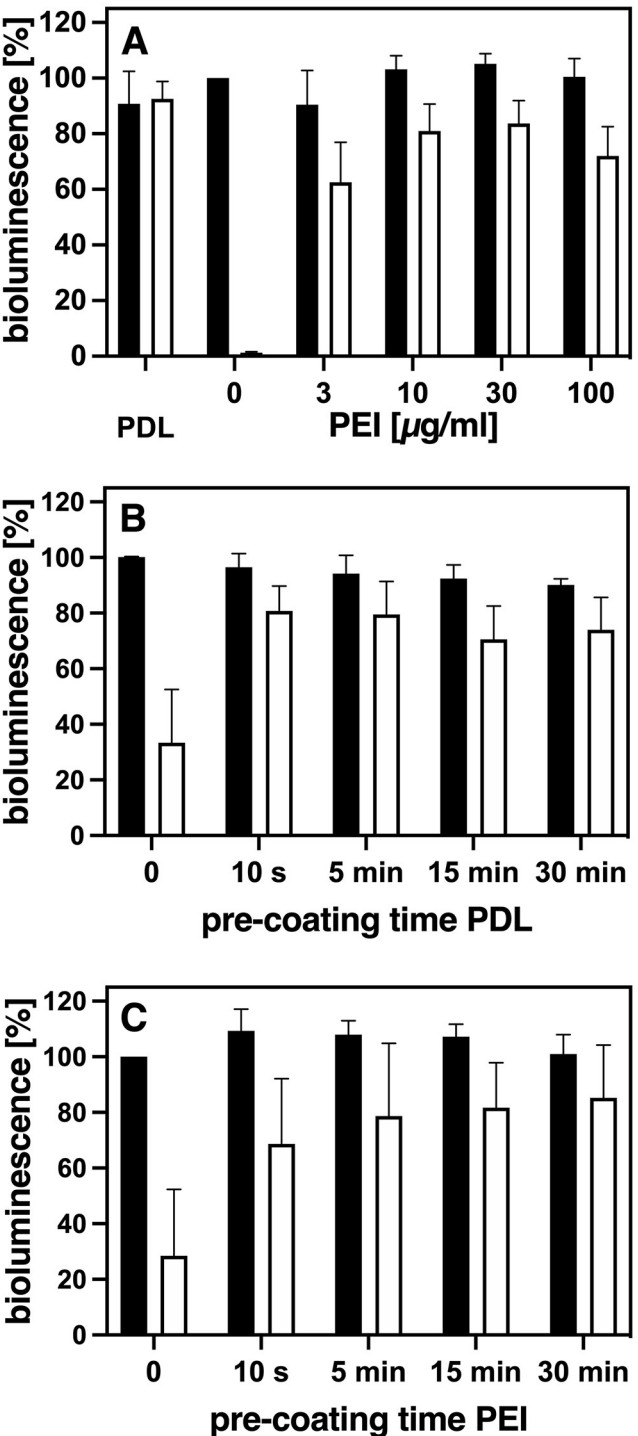

**Fig 2. Pre-coating with increasing PEI concentrations, or with prolonged pre-coating times with either PDL or PEI.** A) Wells were pre-coated either with 100 μl of PDL (100μg/ml) or increasing concentrations of PEI (3–100 μg/ml) for at least 1 h. B) and C) Wells were pre-coated for only 10 s or up to 30 min with either (B) 100 μg/ml PDL or (C) 10 μg/ml PEI. The pre-coating solutions were aspirated and HEK 293 cells expressing either GRPR55- or B$_2$R R128A-RlucII constructs seeded immediately. Strength of cell attachment was estimated after 2–3 d by rinsing the cells with ice-cold PBS using a microplate-washer (white bars) as described in "Materials and methods". For controls (black bars), cell culture medium was removed with a 12-channel pipette without any rinsing of the cells. Bioluminescence was determined as described in "Materials and methods". Columns depict Mean ± SEM (n = 3–6, performed in duplicates). The wells without pre-coating and without rinsing were taken as 100%.

room temperature can be applied. Use of these concentrations requires only aspiration of the pre-coating solutions but no additional rinsing in order to avoid negative effects on the cells seeded afterwards, as indicated by the similar bioluminescence activities of all the control cells without washing (Fig 2A–2C).

An alternative approach to estimate how much cationic polymer left might be tolerated by the cells without rinsing was to test which amounts of them added directly to the medium would affect cell growth and aspect. Quite interestingly, we noticed that with some concentrations of PDL added, the HEK 293 cells not only still grew well but also strongly attached to the culture plates. As direct addition instead of pre-coating would have been a novel, uncomplicated way to promote adherence, we investigated this phenomenon in more detail for both, PDL and PEI.

## PDL or PEI added directly to the medium strongly promote cell adherence

As shown in Fig 3A an addition of PDL at a concentration of 1–5 μg/ml medium resulted in strongly attached cells. Higher PDL concentrations led to less bioluminescence even in the control cells that have not been rinsed. This indicates that PDL concentrations higher than 5 μg/ml medium have a toxic effect and impair the capability of HEK 293 cells to grow and to synthesize functional receptor-RlucII fusion proteins. Equivalent results were obtained for PEI (Fig 3B). Addition of 0.75–3 μg PEI/ml medium resulted in strong anchorage of the cells without affecting their growth or protein synthesis as demonstrated by comparison with the control cells grown in the absence of PEI. However, PEI concentrations higher than 3 μg/ml medium were apparently toxic to the cells. In summary, PDL or PEI at concentrations of 1–5 μg/ml or 0.75–3 μg/ml, respectively, can be added directly to the medium to promote strong adherence of the cells without any notable negative effects on cell growth or protein synthesis.

We, therefore, chose 2 μg/ml medium for PDL and 1.5 μg/ml medium for PEI as standard concentrations in the medium for our experiments.

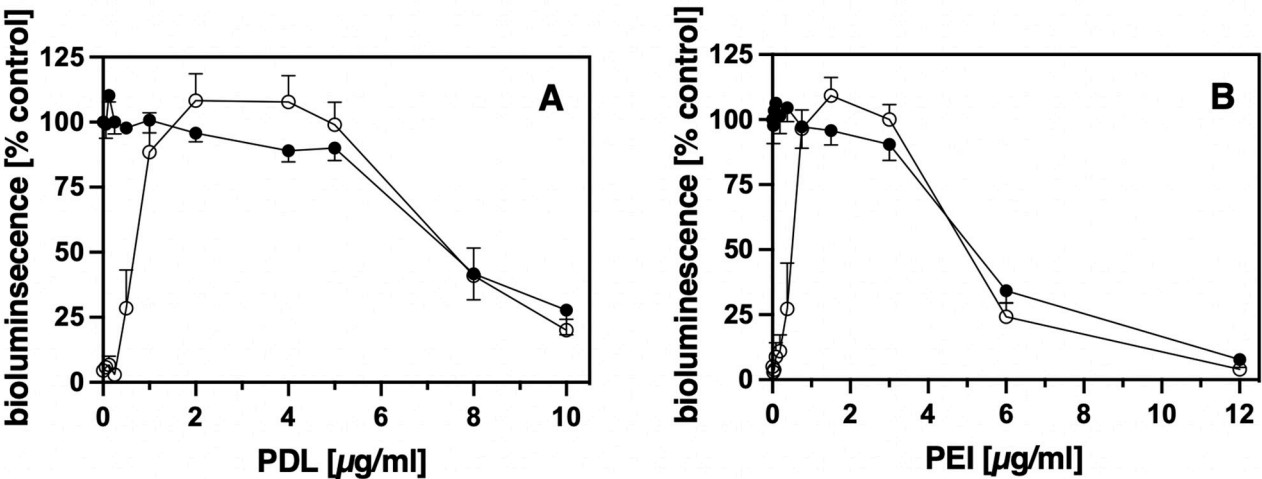

**Fig 3. Direct addition of PDL and PEI to the medium results in strong cell adherence.** HEK 293 cells expressing either GRPR55- or B₂R R128A-RlucII constructs were seeded in complete medium with the addition of increasing concentrations of PDL (A) or PEI (B) as indicated. After 2–3 d strength of cell attachment was estimated as described in the legend of Fig 2. (●) control, (○) rinsed, symbols represent Mean ± SEM (n = 4, performed in duplicates). The wells without addition of cationic polymer and without rinsing were taken as 100%.

## Comparison of pre-coating versus direct addition with regard to cell growth and adhesion

As Fig 4 demonstrates, no differences were observed with regard to cell adherence when PDL or PEI were used in the respective standard concentrations for pre-coating (100 μg/ml or 10 μg/ml, respectively) or by direct addition to the medium (2 μg/ml or 1.5 μg/ml, respectively). Even low amounts of cells seeded grew well under all standard conditions, as there was still an almost linear correlation for all conditions without rinsing challenge between the amounts of cells plated and the bioluminescence measured after two to three days. Moreover, the functionality of the cells was not affected by the method or type of polymer used to achieve cell adherence. Neither release of intracellular calcium after stimulation of the endogenous bradykinin $B_2$ receptor nor of the protease-activated receptor 2 (Par2) nor cAMP synthesis after stimulation of adenylate cyclases by forskolin did differ (S1 Fig). However, the number of the cells seeded mattered when it came to their capability to withstand extensive rinsing. For both polymers and kind of applications (pre-coating or direct addition) a significant loss of cells at lower cell densities could be observed (Fig 4). This is most likely due to lack of stabilization of adherence through cell-cell contacts under these subconfluent conditions.

## Strong cell attachment does not require addition of fetal bovine serum to the medium

Most cells, including HEK 293 cells, do not attach well to cell culture plates without addition of fetal bovine serum (FBS). The question arose whether the cationic polymers strengthen the

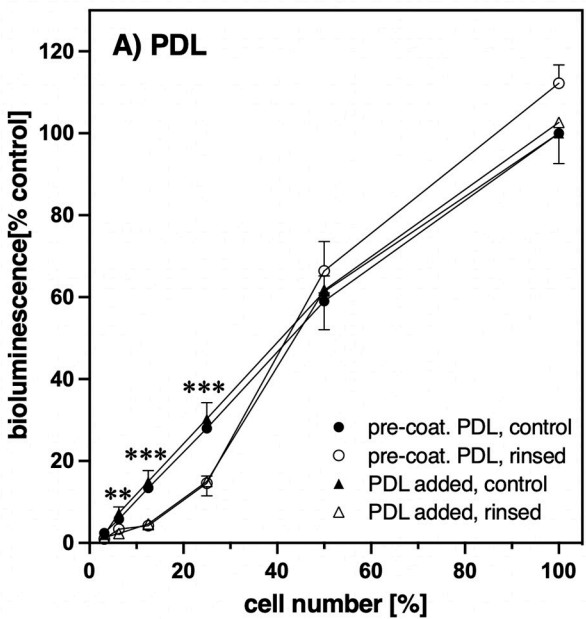
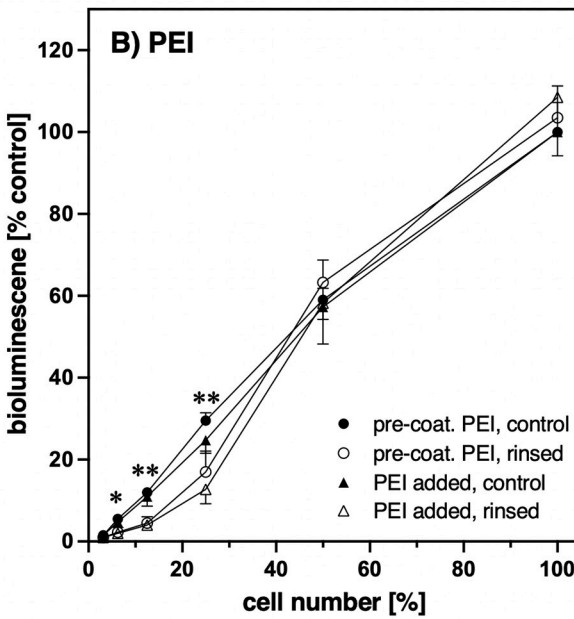

**Fig 4. Effect of pre-coating or of direct addition of PDL and PEI on cell adherence with increasing cell density.** Wells of a 96-well plate were either pre-coated (●, ○) with standard solutions of A) PDL (100 μg/ml) or B) PEI (10 μg/ml) followed by seeding of $B_2R$ R128A-RlucII expressing HEK 293 cells without addition, or the cells were seeded without pre-coating but with addition (▲, △) of A) PDL (2 μg/ml) or PEI (1.5 μg/ml). Cells were serially diluted two-fold starting with approx. 30.000 cells per well. After 2–3 d adherence was determined as described in legend of Fig 2. The wells with the highest cell density were confluent and taken as 100%. Symbols represent Mean ± SEM (n = 4–5, performed in duplicates). Rinsed (●,▲), control (○, △). Multiple unpaired t-test with Welch correction and Holm-Sidak's multiple comparison test for all values (pre-coated and addition together) of rinsed vs. control cells for each dilution: *p<0.05, **p<0.01, ***p<0.001.

cell attachment by themselves or work by recruiting factors, such as e.g. fibronectin or vitronectin, from the FBS to the cells and the plastic surface. To address this question, we tested different overnight pre-treatment conditions of the wells: we used medium with or without addition of the cationic polymers, combined or not with addition of FBS (Fig 5, column pairs and images a-f). Moreover, we pre-coated wells with standard concentrations of PDL or PEI for 1 h followed by overnight incubation with medium containing FBS (Fig 5, column pairs and images g, h). The next day cells were seeded in medium without any addition of polymer or of FBS (Fig 5, column pairs and images a-h). This experimental set-up should clarify whether the cationic polymers (added or used for pre-coating) can recruit factors from the FBS that would suffice for mediating strong attachment of cells seeded without any additional supplement of FBS or of cationic polymer. As further controls, we studied the attachment when cells were seeded with standard concentrations of the cationic polymers in the presence and absence of FBS (Fig 5, column pairs and images i-l). All cell incubation was done only overnight as HEK 293 cells without addition of FBS hardly grow and survive for more than one day. Moreover, these experiments were done with two different cell densities: a lower one that under normal circumstances would result the next day in a subconfluent monolayer (seeding densities < 50.000 cells/well, Fig 5A and 5C) and a higher one that would give a confluent monolayer (seeding densities > 80.000 cells/well, Fig 5B and 5D).

When the cells were seeded in medium without any supplement in wells that were pre-treated overnight with medium without or with FBS only, most of the cells somehow attached themselves to the well bottom (Fig 5C and 5D, images a and b, respectively). Their adherence, however, was very weak as they could not withstand any rinsing challenge (Fig 5A and 5B, white columns of pairs a and b, respectively). Unexpectedly however, overnight pre-treatment of wells with medium containing only cationic polymers but no FBS—originally designed as negative control for the same pre-treatment with FBS—was enough to promote strong adherence of HEK 293 cells seeded afterwards in medium alone (Fig 5A–5D, column pairs and images c and d). Addition of FBS to the overnight pre-treatment with the cationic polymers (Fig 5A–5D, columns and images e and f) could not further improve cell adherence. The same was observed with a 1h pre-treatment with cationic polymers (pre-PDL, pre-PEI) followed by overnight incubation with medium with FBS (Fig 5A–5D, column pairs and images g and h). FBS, however, was apparently important to protect the cells from potential damage by addition of the cationic polymers. When the cells were incubated overnight in medium with cationic polymers in the absence of any FBS, they did not attach well, were much less resistant to rinsing, and a lot of cell debris could be seen (Fig 5A–5D, column pairs and images i and j) as compared to the standard incubation with addition of FBS (Fig 5A–5D, column pairs and images k and l). This effect was particularly observed with lower cell densities (Fig 5A and 5C), and PDL was more harmful than PEI in this context. These studies demonstrate that, interestingly, FBS is not required for strong cell adherence when the culture plates are pre-coated with cationic polymers. Moreover, cell-cell-interactions contribute considerably to the stability of cell adherence, as independent of the kind of treatment, relatively more cells of a subconfluent monolayer were lost with rinsing than of a confluent one (Fig 5A and 5B, respectively).

An additional indication that not specific factors from the FBS are pivotal for strong cell adhesion comes from experiments with animal-free cell culture conditions. Pre-coating with the standard concentrations of PDL or PEI or addition of them also worked comparably well when FBS was replaced by 10% human platelet lysate, provided that—against the recommendation of the manufacturer—no heparin is added as anti-coagulant to the culture medium (S2 and S3 Figs). Addition of the negatively charged heparin is apparently not needed for cultivation of the HEK 293 cells but clearly prevents strong attachment of these cells when using the cationic polymers.

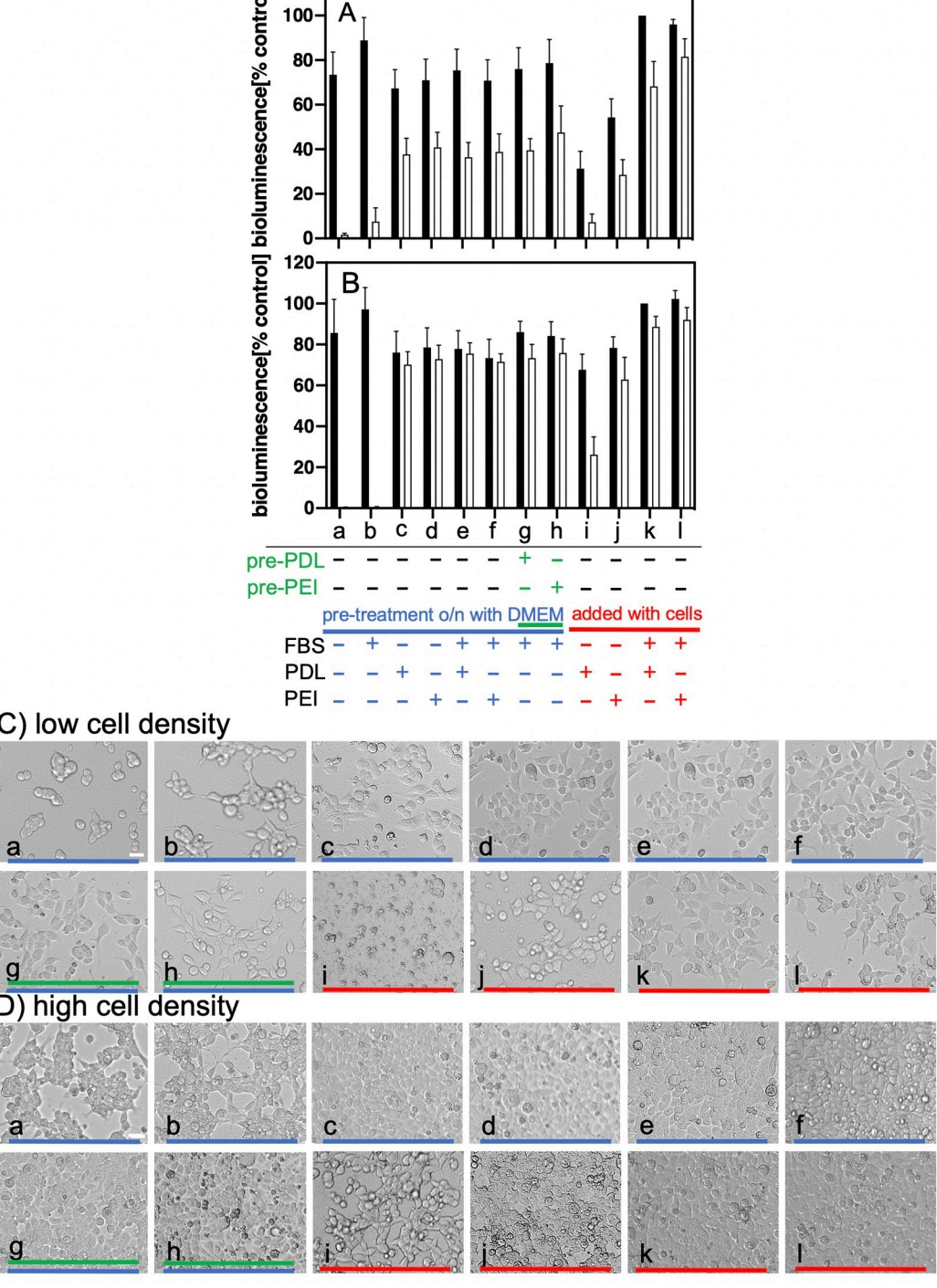

**Fig 5. Fetal bovine serum not required for strong cell adhesion after PDL or PEI pre-treatment.** Wells were pre-treated overnight with medium without or with addition of fetal bovine serum (FBS), PDL (2 µg/ml), or PEI (1.5 µg/ml) as indicated. Some wells were first pre-treated 1 h with either PDL (100µg/ml) or PEI (10µg/ml) (pre-PDL, pre-PEI; green bars). After 16 h cells were seeded either in medium **only** (a-h; blue bars) or in medium with addition of FBS, PDL, or PEI as indicated (i-l; red bars). The next day, cell attachment was determined as described in "Materials and methods". A) and C), cell densities < 50.000 cells/well (subconfluent next day); B) and D), cell densities > 80.000 cells/well (confluent next day). The columns represent Mean ± SEM (n ≥ 6, performed in duplicates); (black) control, (white) rinsed. The wells with addition of FBS and PDL but without rinsing were taken as 100%. C) and D) representative phase-contrast images taken for low and high cell densities, respectively (white bar = 20 µm). Small letters and coloured bars refer to the column pairs and their conditions as outlined below Fig A and B.

Another semi-adherent cell line that is quite important in cell culture research is the pheochromocytoma-derived PC-12 cell line [13]. Addition of or pre-coating with our standard concentrations of PDL or PEI also worked comparably well for this cell line. The PC-12 cells seeded under our standard conditions on a 24-well plate withstood careful, gentle rinsing without distinct loss (S4 Fig). However, when challenged in the adherence test, only pre-coating, in particular with PEI, prevented the loss of cells, whereas addition of the cationic polymers did not (S5 Fig). these results suggest that for each cell line specific conditions have to be determined for promoting strong cell adherence.

## Discussion

Cells in culture are differently strongly attached to the culture plates, even if they are grown on cell culture grade plastic, depending on cell type, passage, origin, and composition of the cell culture medium.

Many experiments require specific handling that can include rinsing of the cell monolayers several times, fast changes in incubation temperature, or rapid addition of reagents which might generate sheer forces. Moreover, stimulation of e.g. overexpressed $G\alpha_{q/11}$-coupled GPCRs results in rapid intracellular increase of $Ca^{2+}$ with strong cell contractions and thus additional weakening of cell anchoring [3]. The most commonly, almost exclusively used method to promote the adherence of otherwise weakly attached cells such as HEK 293 or PC-12 cells, devised almost 45 years ago, is to pre-coat the culture plates with poly-L/D-lysine (PLL/PDL) [1]. Other approaches reported, but never widely applied, were pre-coating with other cationic polymers such as poly-ethylenimine [12], or poly-ornithine [14], or even the overexpression of class A macrophage scavenger receptors (SR-A1) in HEK 293 cells [15]. As the latter can bind to poly-anionic ligands it presumably mediates the anchoring of the HEK 293 cells by binding directly to the negatively charged surfaces of culture plates. Differential natural expression of membrane proteins with similar properties as SR-A1 might in part explain the different attachment strength of cells.

We set out to develop a well-defined, possibly simplified PDL pre-coating protocol, as the existing ones did not elucidate in detail how the various parameters, such as concentrations, time, need for rinsing and drying, were obtained or justified, and whether they also worked for subconfluent cell densities. The latter is most likely due to the fact, that there were no suitable approaches to identify reliably lower amounts of poorly attached cells in a well and thus these protocols were developed mostly for confluent monolayers. For our approach, we combined a very sensitive bioluminescence method to estimate the amount of cells with a challenging protocol to assess the adherence strength of cells, in our case HEK 293 or PC-12 cells.

The magnitude of bioluminescence measured correlates well after 2–3 days with the relative amount of HEK 293 cells seeded that stably express GPCR-RlucII fusion constructs (Fig 1). The bioluminescence measured was not significantly affected by whether the cells were strongly attached through PDL pre-coating of the wells or not. This measurement requires only one exchange of culture medium for HBSS, but does not involve any additional fixation or washing steps as e.g. needed for staining methods [12]. Such additional steps might result in loss of cells, thus distorting the data. As the bioluminescence approach is very sensitive, even fairly low amounts of cells can easily and reliably be quantified (Figs 1 and 4). To assess the anchoring strength of the HEK 293 cells, they were subjected to a challenging protocol by placing the plates on ice and rinsing the cells with ice-cold buffer six times within 15 min using a microplate-washer. Without pre-coating, almost all cells are lost under these conditions (Fig 2A), whereas using the standard pre-coating procedure with PDL (100 μg/ml) 80% and more cells of a confluent monolayer can withstand the rinsing challenge (Fig 2A). These results

demonstrate that we had the right tools to investigate the optimal cell adhesion conditions, even for subconfluent cells.

Our results obtained, applying this approach, show that PDL and the less expensive PEI both can be used for pre-coating. It takes only 5 min incubation time with the respective standard concentrations (100 μg/ml for PDL and 10 μg/ml for PEI) followed by removal of the pre-coating solution without any additional rinsing to obtain strong cell adherence (Fig 2). Use of higher concentrations would require rinsing of the plates before seeding or else remaining pre-coating solutions could harm the cells seeded afterwards. The detrimental effect of higher concentrations of the cationic polymers can also be clearly seen in Fig 3, where higher PDL or PEI concentrations result in a strong reduction of bioluminescence even in the respective unwashed control cells.

Most importantly and novel, however, we found that any pre-coating protocol could be replaced by direct addition of the cationic polymers (2μg/ml for PDL, and 1.5 μg/ml for PEI) to the cell suspension before seeding (Fig 3 for HEK 293 cells and in part for PC-12 cells S4 Fig). The cells with standard polymer addition behave identically to cells on pre-coated wells (Fig 4). However, with PDL or PEI added to the medium also fetal bovine serum (FBS) is required as otherwise the cationic polymers are not tolerated well by the cells (Fig 5). With higher concentrations of the cationic polymers even the presence of FBS cannot protect the cells as indicated by the reduced bioluminescence of the respective control cells in Fig 3. In spite of this, our results also show that the addition of fetal bovine serum is not needed for firm anchoring of the cells as long as the culture plates were pre-treated with cationic polymers (Fig 5). Although in general a cooperative recruitment of factors like fibronectin or vitronectin from FBS by the cationic polymers cannot be excluded, these factors are obviously not essential for strong cell attachment. Therefore, e.g. for fast serum starvation assays, it would be possible to get strongly adhering HEK 293 cells on PDL or PEI pre-coated cell culture plates without addition of serum.

Using our combination of sensitive bioluminescence measurement with the rinsing challenge, we can show that cells growing at lower densities are more susceptible to detachment when rinsed than confluent monolayers, independent of the (pre-)treatments (Figs 4 and 5). This is presumably due to the lack of the stabilizing effect of an increased number of cell-cell-interactions with higher cell densities. As most experimental handling requires distinctly less rigorous treatment than our rinsing challenge, pre-coating or a direct addition of PDL or PEI to the medium should in general result in even less loss of cells than observed with our subconfluent examples (Figs 4 and 5B, and S4 Fig).

Nevo et al. studied 65 years ago the agglutination of erythrocytes through addition of cationic polymers [16]. They observed that depending on the size and quantity of the polymer the electrophoretic mobility of the cells shifted from negative to positive, as these polymers attached themselves to the negatively charged surfaces of the erythrocytes. Overcoming thus the repulsion of the negative cell surfaces, the positively charged polymers started the agglutination of the erythrocytes. Similarly, the positively charged polymers PDL and PEI could function by eliminating the repulsion between the negatively charged cell culture plate and the negatively charged cell surface. In support of this mechanism PDL and PEI work similarly well for pre-coating or when added directly, despite having quite a different composition. Apparently, for promoting cell adhesion of cells like HEK 293 or PC-12 cells, the polymer must only display a high density of positive charges at neutral pH, thus explaining why e.g. also a polymer like poly-ornithine could be used for pre-coating purposes [14].

Weakly attached semi-adherent cells presumably do not produce enough extracellular matrix or do not have sufficient suitable membrane proteins such as the SR-A1 that would help them attach stronger to the plastic surface of cell culture plates that were negatively

charged through exposure to plasma gas. There are some experimental advantages of such reduced adherence as e.g. HEK 293 cells can be easily detached by solutions containing just $Ca^{2+}/Mg^{2+}$-chelators such as EDTA. Moreover, we observed that HEK 293 cells can be seeded several times in the same cell culture plates and flasks without any adverse effect on their growth, appearance or adherence, whereas other cells, e.g. strongly anchoring human skin fibroblasts, do not tolerate being seeded on a recycled plate for a second time (personal observation). However, for most experiments strong cell attachment is of great advantage, if not essential.

Here, we could report that the cumbersome, time-consuming pre-coating can be replaced by a simple, but novel method, the direct addition of the basic polymers to the medium when seeding the HEK 293 cells and with some limitation PC-12 cells, without any harmful side effects on their growth or appearance. With the increasing utilization of 384-well plates this ease of handling might become even more advantageous.

## Supporting information

**S1 Table. Comparison of luminescence measured in white and black plates with application of white stickers for both types.** Four different concentrations of HEK 293 cells stably expressing the B2 R128A-RlucII construct were seeded identically either in white (top) or in black (bottom) 96-well plates in the wells labeled yellow. After two days, a white sticker was attached to the bottom of the plates and bioluminescence determined as described in "Materials and methods" in all wells (empty or with cells). Note: HBSS/HEPES and coelenterazine H was added only to the wells marked yellow. On the left side are given the absolute values, on the right side the values as percentage of the respective well containing the cells (marked yellow). To keep interference at a minimum only the respective parts of the plates shown were measured at a time, starting with the lower bioluminescence in row H. The results show that luminescence can be measure in black plates with a white sticker, resulting in lower values than in white plates but also distinctly less interference between the wells.
(TIF)

**S1 Fig. Signalling functionality of cells is independent of how adherence was accomplished.** HEK 293 cells with stable integration of the 20F cAMP plasmid (Promega) for measurement of changes of the intracellular cAMP level were seeded in 96- or 384-wells either pre-coated with PDL (green curves) or with addition of PDL (red curves) or PEI (blue curves) to the medium. After 2–3 days, the respective measurements were performed in HBSS/ HEPES. Left and middle panel: The FLPR calcium 5 Assay kit (Molecular Devices) was used to measure intracellular calcium release according to the instructions of the manufacturer. Cells (in black 384 wells with clear bottom) were stimulated at 37˚C by injector addition of bradykinin (BK) or Par2 agonist (2-Furoyl-LIGRLO-NH$_2$), both with 1 μM final concentration, as indicated. Fluorescence was measured in a Tecan Infinite F200 PRO microplate reader (excitation: 485 nm, emission 525 nm). Symbols in the middle panel represent Mean ± SD of one measurement in duplicates. Right panel: Cells were seeded in a white 96-well plate with flat, clear bottom (PerkinElmer) as described in "Materials and methods". After application of a white sticker to the bottom of the plate, luminescence reflecting cAMP synthesis was measured in the Tecan microplate reader at 23˚C after addition of luciferin-EF (Promega; 3 μM, 67.5 μl/well, 90 min pre-incubation) and stimulation of adenylate cyclase activity with forskolin (Sigma, 1 μM final concentration, added in 7.5 μl) as indicated. Symbols represent mean ± SD of one measurement in quadruplets. The results of these stimulations of the endogenously expressed BK B2 or the Par2 receptors and of stimulation of

adenylate cyclase activity did not differ dependent on the method or cationic polymer used to obtained cell adherence.
(TIF)

**S2 Fig. Comparison of the effect of PDL and PEI on the adherence of HEK 293 cells seeded in FBS-containing and animal-free cell culture media.** HEK 293 cells expressing either GRPR55- or B$_2$R R128A-RlucII constructs were detached with trypsin and centrifuged (100xg, 5 min) after addition of 5 ml complete medium. The resulting cell pellet was resuspended in 700 μl DMEM without any supplements. 200 μl each thereof were added to three wells of a 6-well plate containing 7 ml of a) complete medium (**FBS-medium**), b) DMEM + 10% human platelet lysate (ELAREM Prime, PL BioScience) with 2U/ml heparin (sodium salt, Serva) (**hPL-medium+heparin**) or c) hPL-medium without addition of heparin (**hPL-medium w/o heparin**). All cell suspensions contained 1% penicillin/streptomycin and tetracycline (0.5 μg/ml) for induction of expression of the RlucII constructs. Twice 1.5 ml of each cell suspension was transferred to two wells of a 24-well plate and either PDL (2 μg/ml final conc.) or PEI (1.5 μg/ml final conc.) added. A 12-channel pipette with a suitable number of tips was used to transfer 200 μl of each cell suspension from either the 6-well or the 24-well plate to a black 96-well plate with wells pre-treated or not as indicated with PDL (100 μg/ml; **pre-PDL**) or PEI (10 μg/ml; **pre-PEI**). After 2 days cell adherence was estimated by washing the cells with ice-cold PBS using a microplate-washer (white bars) as described in "Materials and methods". For controls (black bars), cell culture medium was removed with a 12-channel pipette without any washing of the cells. Bioluminescence was determined as described in "Materials and methods". The wells seeded in FBS-medium without pre-coating or additions, and without washing, were taken as 100% for all other conditions. Columns depict Mean ± SEM (n = 4, performed in triplicates, controls in quadruplets). S4 Fig shows that cells seeded in hPL-medium performed with regard to adherence comparable to those seeded in FBS-medium provided that no heparin had been added. This negative charged polymer apparently prevents the adherence promoting effect of the cationic polymers PDL and PEI, at least with the concentrations used here.
(TIF)

**S3 Fig. Comparison of the effect of PDL and PEI on the adherence of HEK 293 cells seeded in FBS containing and animal-free cell culture media: phase-contrast-images.** Representative phase-contrast-images images (10x) from the experiment presented in S2 Fig. S3 Fig (top) shows that with same amounts of HEK 293 cells added, after 2 days with FBS-medium confluent monolayers were obtained independent of the pre-treatment of the wells or addition of cationic polymers. When FBS was exchanged for hPL (human platelet lysate), in the presences of heparin under no conditions the cells became confluent (middle), whereas in the absence of heparin an almost confluent monolayer was generated either when the wells were pre-treated or when PDL or PEI was added to the medium (bottom). However, in the control cells even in the absence of heparin, the cells remained subconfluent and a had a less flattened appearance as observed for all cells in the presence of heparin.
(TIF)

**S4 Fig. Effect of PDL/PEI pre-coating or PDL/PEI addition on growth and adherence of PC-12 cells I.** PC-12 cells: ATCC CRL-1721. Cultivation medium: RPMI medium + 15% horse serum + 2.5% fetal bovine serum + 1% penicillin/streptomycin (all from Sigma). Cells were split 1:2 (high confluency) or 1:4 (low confluency) without trypsin and seeded in 24-wells in 800 μl or 400 μl medium, respectively, as indicated either in wells pre-treated with PDL (100 μg/ml) or PEI (10 μg/ml), or with addition of PDL (2 μg/ml) and PEI (1.5 μg/ml). After 2

days phase-contrast-images (white bar = 20 μm) were taken of the cells without (one image each for high and low confluency) and after gentle rinsing twice with 0.5 ml PBS (two images each for only low confluency). PC-12 cells were growing well attached after pre-coating with PDL or PEI, or with addition of the PEI as compared to no pre-coating or no addition. They grew less well with addition of PDL. After pre-coating with PDL or PEI, or PEI addition they withstood moderate rinsing with PBS without considerable loss of cells. This was not the case with cells seeded without pre-coating or any addition. There, almost all cells were lost. Cells grown with addition of PDL were apparently also less well attached and displayed some loss with rinsing.
(TIF)

**S5 Fig. Effect of PDL/PEI pre-coating or PDL/PEI addition on growth and adherence of PC-12 cells II.** Cultivation medium: RPMI medium + 15% horse serum + 2.5% fetal calf serum + 1% penicillin/streptomycin (all from Sigma). Cells were split 1:4 without trypsin and seeded in wells of a 96-well in 200 μl medium, as indicated either in wells pre-treated with PDL (100 μg/ml) or PEI (10μg/ml), or with direct addition of PDL (2 μg/ml) or PEI (1.5 μl/ml) to the medium. After 2 days phase-contrast-images (white bar = 40 μm) were taken of the cells before (four images each) and after (two images each) performing the adherence test as described in "Materials and methods". PC-12 cells were growing well attached after pre-coating with PDL or PEI, or addition of PEI as compared to no pre-coating or additions. With PDL or PEI pre-coating the cells withstood the adherence test without considerable loss. In contrast, PC-12 cells grown with addition of PEI displayed considerable reduction of cells and those grown in the presence of PDL were almost completely lost, comparable to the control cells without pre-coating or additions.
(TIF)

## Author Contributions

**Conceptualization:** Alexander Faussner.

**Investigation:** Alexander Faussner, Matthias M. Deininger.

**Methodology:** Alexander Faussner, Matthias M. Deininger.

**Writing – original draft:** Alexander Faussner, Matthias M. Deininger.

**Writing – review & editing:** Alexander Faussner, Matthias M. Deininger, Christian Weber, Sabine Steffens.

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
