## [Decision Letter · Decision Letter 0]

19 Dec 2021

PONE-D-21-34583Direct addition of poly-lysine or poly-ethylenimine to the medium, a simple alternative to plate pre-coatingPLOS ONE

Dear Dr. Faussner,

Thank you for submitting your manuscript to PLOS ONE. After careful consideration, we feel that it has merit but does not fully meet PLOS ONE’s publication criteria as it currently stands. Therefore, we invite you to submit a revised version of the manuscript that addresses the points raised during the review process.

We look forward to receiving your revised manuscript.

Kind regards,

Nazmul Haque

Academic Editor

PLOS ONE

Journal Requirements:

This work was supported by a grant from Foerderprogramm fuer Forschung und Lehre der LMU Munich (FoeFoLe; to A.F. and M.D.)

This work was supported by a grant from Foerderprogramm fuer Forschung und Lehre der LMU Munich (FöFoLe; to A.F. and M. D.)

https://www.med.uni-muenchen.de/forschung/foerderprogramme/foefole/index.html

Reviewers' comments:

Reviewer's Responses to Questions

**Comments to the Author**

1. Is the manuscript technically sound, and do the data support the conclusions?

Reviewer #1: Yes

Reviewer #2: Partly

2. Has the statistical analysis been performed appropriately and rigorously? 

Reviewer #1: Yes

Reviewer #2: Yes

3. Have the authors made all data underlying the findings in their manuscript fully available?

Reviewer #1: Yes

Reviewer #2: Yes

4. Is the manuscript presented in an intelligible fashion and written in standard English?

Reviewer #1: Yes

Reviewer #2: Yes

5. Review Comments to the Author

Reviewer #1: The manuscript submitted by Alexander Faussner et al simplified the current method to promote strong cell attachment by plate pre-coating with PDL and demonstrated that the PEI can be used in the method as well. More importantly, the authors reported a novel method, that is, direct addition of the cationic polymer into the culture medium, which results in comparable strong cell anchorage as the standard plate pre-coating method. The new method is highly labour-, time-, and cost-efficient as compared to the current method. Overall, their experimental design, data analysis, and results are sound, and I have no major objections for its publication in our journal if following questions have been addressed.

Followings are a few questions for the authors.

1. Have the authors tested the direct addition method in other semi-adherent cell lines?

2. Whether pre-coated with other cationic polymer (e.g., PEI) or using direct addition method or under different coating/addition conditions would impact the correlation between the cell density and bioluminescence? Any data to support?

3. In figure 3, is there any explanation on why the rinsed conditions in the optimum concentration range for both cationic polymers, especially PDL, are consistently higher than control?

4. Is there any further investigation into the protective effects of FBS in direct addition method (e.g., contributing components and minimal FBS concentration required)?

Reviewer #2: Manuscript ID: PONE-D-21-34583

The manuscript entitled "Direct addition of poly-lysine or poly-ethylenimine to the medium, a simple alternative to plate pre-coating" is focused on a topic of paramount interest in veterinary medicine and there is a strong linking with human field.

In general, the manuscript needs revisions with a particular regard to the study design and the presentation of results. The data are very confused; Materials and Methods do not describe the procedures applied to perform the study and the chapter Results is written with a lot of poorly organized information.

In particular, it seems that there is not a clear scheme in the construction of the paper.

Introduction

In general, in all the chapter it would be useful to revise the reference: line 52….., line 59

Line 44….. It lacks reference

Line 50 The full name of HEK 293 should be entered

Line 63 Is this the aim of the study? It should be better highlighted

Materials and methods

It might be useful to write few lines of introduction to experimental design.

Cell culture

As aforementioned, the experimental design is not well described.

Line 103 What is the source of cell culture? Are they checked for quality standard? Microbiological and virological control, etc?

Some reagents (line 107, 111) lack the reference to the manufacturer.

Line 108 What was the percentage of vitality? It is necessary to better describe the procedure and the data related to the cells. The information of line 115 is not sufficient.

Line 109-110 It is confused. The application of the procedure is not clear.

Line 116 “30.000 cells per 96-well”, but how many per well? And how many replicates?

Adherence test

Check the reference to the manufacturer

Line 120 “the 96-well tray was put on ice”. Not in a freezer?

Results

As aforementioned, the chapter is poorly organized. It should be rewritten with a more detailed and accurate scheme.

It is really challenging to follow the presentation of the results, because there are not a clear outline of the tests performed.

Line 161 “2-3 days”. The time should be defined or 2 or 3 days, better 48 or 72 hours. The evaluation should be performed at 24 or 72 hours, with no margin of approximation, as cell growth and cell adhesion can vary greatly within this time frame.

Line 184 the correct title of the chapter is Materials and Methods. …. “duplicates”, this the first time that the Authors describe the study design. So, was the experiment run in duplicate? How many plates per test?

Line 228 Is this a title?

Line 262 “n=4-5, performed in duplicates”, what does it mean? In Fig 3 the authors write n=4. Where is it written what these numbers refer to?

Line 286-287 How did the authors calculate the % of cells to seed in each well? How did the authors say that 50000 is “a lower one that under normal circumstances” and 80000 is “a higher one that…”. In legend of Fig 1 the authors write that they seeded 30000 cells/96 wells. Is it possible to have a justification fo the choice of these numbers?

Line 321 Therefore? What could be the actual usefulness and feasibility of precoating a plate with cationic polymers in relation to the non use of SFB? The authors stae that in any case, serum is needed for cells growing. Would not it have been more useful to test with an artificial serum? In this way, it could have been said that system can be considered as animal-free model.

Figure and graphics.

The graphics are not clear: symbols overlap and it is very difficult to have a clear view of the data (fig 3A and B, fig 4 A and B).

Fig 4, the legend is not clear. Fig 5 is not clear, there are too much data and information to link each other.

 

Discussion

The discussion should be more incisive and fitting with the aim of the study.

Line 149 “without any harmful side effects on their growth or phenotype”. On what basis do the authors build this statement? They did not perform a DNA mitochondrial test or soft agar assay to evaluate possible changes in the features of the cells. Maybe it would be better to rephrase the sentence.

Line 372 It would be better to explain what is “clearly seen in Fig 3”

Line 419 What is the basis to affirm this? What is the test performed to establish that the cells have not undergone nay changes in phenotype? The observation by microscope?

Line 420 A concluding sentence is needed, which summarizes what has been highlighted and which is fitting with the aim of the paper.

6. PLOS authors have the option to publish the peer review history of their article (what does this mean?). If published, this will include your full peer review and any attached files.

Reviewer #1: **Yes: **Mian Huang

Reviewer #2: No

---

## [Author Response · Author response to Decision Letter 0]

31 Mar 2022

Response to reviewers: 

 We want to thank the reviewers for their effort and input that motivated us and gave us additional ideas and resulted in a manuscript that in our opinion is now considerably improved.

Reviewer 1: 

 Reviewer 1 raised several interesting points that we tried to address as thoroughly as possible.

Point 1: Have the authors tested the direct addition method in other semi-adherent cell lines?

 Following up on the reviewer’s question, we have tested the same conditions on PC-12 cells that are growing semi-adherently and added two figures in the supporting information section. The images show that pre-coating with either PDL or PEI strongly increases adherence of these cells, however, only addition of PEI results in strong adherence but not that of PDL. Thus, for direct addition, the working concentration of PEI or PDL might have to be determined individually for each semi-adherent cell line. Text dealing with this topic has been added to the manuscript.

Point 2: Whether pre-coated with other cationic polymer (e.g., PEI) or using direct addition method or under different coating/addition conditions would impact the correlation between the cell density and bioluminescence? Any data to support?

 To address this question, we changed Figure 1 to show in a direct comparison the effect of pre-coating or lack thereof on the bioluminescence signal. There is apparently a trend for stronger bioluminescence in untreated wells as compared to pre-coated ones that however is not significant. Moreover, as figure 3 indicates, there is no difference between the bioluminescence measured in cells with low adherence vs. that measured in strongly attached cells that can withstand the rinsing challenge.

Point 3: 

In figure 3, is there any explanation on why the rinsed conditions in the optimum concentration range for both cationic polymers, especially PDL, are consistently higher than control?

 We noticed this of course, too. However, as the data for rinsed in question were not significantly different (multiple t-test) from the respective data for controls, we did not consider this as relevant. Indeed, in most of the other experiments it's the other way around. It could be an edge effect, because the control cells were located in row A that sometimes has a somewhat lower cell density despite have received the same amounts of cells (at least in theory). For following experiments we alternated.

Point 4:Is there any further investigation into the protective effects of FBS in direct addition method (e.g., contributing components and minimal FBS concentration required)?

 This is an interesting question, too, but was not the main topic of this study. One might speculate that as higher concentrations of PDL or PEI are harmful to the cells even in the presence of FBS, that without FBS lower concentrations of PDL or PEI should be used and might be enough to promote cell adherence. We tested whether in the absence of FBS lower amounts of PDL or PEI would allow the cells to grow healthy and still well attached. As this however, disappointingly, was not the case (with lower concentrations of PDL/PEI the cells were not well attached) we did not continue in that direction. 

Reviewer 2:

 We are sorry that we obviously gave reviewer 2 a hard time with reading our manuscript. In the revised version we tried to improve its legibility by rephrasing significant parts of it and addressing the concerns raised in this thorough and detailed review. We want to thank him for pointing out that this topic is of high importance to the veterinary field as we were not aware of this and actually would like to know more about it.

General response:

Introduction: 

We entered the full name of the HEK293 cells and stated the aim of our study in more detail

Materials and methods: 

We enter the lacking references, corrected inaccuracies and tried to describe the procedures in a less confusing way, giving more information and explaining the methods in more detail (in part also entering relating details in the supporting material section).

Line 103:

The source of our cells was given in line 102 and 104. The cells were checked by vision for health and viability. A mycoplasma test was once performed and found negative. 

As hardly any cell were floating the next day after seeding under normal conditions it was assumed that they are in principle all viable. Of course, there were experiments with cells that did not look healthy e.g. when plated without FBS in the presence of PDL or PEI. But these are conditions that are not to be recommended anyway.

Results:

We hope that with the changes and additions in the “Materials and methods” as well with the new supporting information it is now more evident how we designed and performed our experiments.

More detailed responses:

Line 161 “2-3 days”. The time should be defined or 2 or 3 days, better 48 or 72 hours. The evaluation should be performed at 24 or 72 hours, with no margin of approximation, as cell growth and cell adhesion can vary greatly within this time frame.

Indeed, the cells are not always growing the same way that’s why we gave this time frame of 2-3 days. The effect e.g. of different lots of FBS is not negligible. But the fact that despite this the STDs are acceptable demonstrates the robustness of the results and the assay. 

Line 286-287 How did the authors calculate the % of cells to seed in each well? How did the authors say that 50000 is “a lower one that under normal circumstances” and 80000 is “a higher one that…”. In legend of Fig 1 the authors write that they seeded 30000 cells/96 wells. Is it possible to have a justification fo the choice of these numbers?

This included various experiments with different non-standardized numbers of cells. From these experiments we learned that it takes more than 80.000 cells/well for the next day to be confluent, otherwise they are subconfluent. For the experiments with a measurement after 2-3 days in general approximately 30.000 cells/well were used as described to obtain confluent monolayers after 2-3 days.

Line 321 Therefore? What could be the actual usefulness and feasibility of precoating a plate with cationic polymers in relation to the non use of SFB? The authors stae that in any case, serum is needed for cells growing. Would not it have been more useful to test with an artificial serum? In this way, it could have been said that system can be considered as animal-free model.

As written in the manuscript the pre-coating without FBS in the medium afterwards was intended as a negative control that turned out to give an unexpected result.

Following the suggestion of the reviewer we added new, in our opinion interesting experiments in the supporting informations with animal-free medium where the FBS is substituted with human platelet lysate.

Figure and graphics.

The graphics are not clear: symbols overlap and it is very difficult to have a clear view of the data (fig 3A and B, fig 4 A and B).

 We reduced the symbol sizes and use now one-sided SEM-bars.

Fig 4, the legend is not clear. Fig 5 is not clear, there are too much data and information to link each other.

 Fig 4 We tried to make the legend more intelligible

 Fig. 5 We changed the figure design and added color

---

## [Decision Letter · Decision Letter 1]

1 May 2022

Direct addition of poly-lysine or poly-ethylenimine to the medium: a simple alternative to plate pre-coating

PONE-D-21-34583R1

Dear Dr. Faussner,

We’re pleased to inform you that your manuscript has been judged scientifically suitable for publication and will be formally accepted for publication once it meets all outstanding technical requirements.

Kind regards,

Nazmul Haque

Academic Editor

PLOS ONE

Additional Editor Comments (optional):

Reviewers' comments:

Reviewer's Responses to Questions

**Comments to the Author**

1. If the authors have adequately addressed your comments raised in a previous round of review and you feel that this manuscript is now acceptable for publication, you may indicate that here to bypass the “Comments to the Author” section, enter your conflict of interest statement in the “Confidential to Editor” section, and submit your "Accept" recommendation.

Reviewer #1: All comments have been addressed

Reviewer #2: All comments have been addressed

2. Is the manuscript technically sound, and do the data support the conclusions?

Reviewer #1: (No Response)

Reviewer #2: Yes

3. Has the statistical analysis been performed appropriately and rigorously? 

Reviewer #1: (No Response)

Reviewer #2: Yes

4. Have the authors made all data underlying the findings in their manuscript fully available?

Reviewer #1: (No Response)

Reviewer #2: Yes

5. Is the manuscript presented in an intelligible fashion and written in standard English?

Reviewer #1: (No Response)

Reviewer #2: Yes

6. Review Comments to the Author

Reviewer #1: (No Response)

Reviewer #2: The authors responded appropriately to the comments presented. In addition, they implemented the paper with additional information that makes the data more exhaustive.

7. PLOS authors have the option to publish the peer review history of their article (what does this mean?). If published, this will include your full peer review and any attached files.

Reviewer #1: **Yes: **Mian Huang

Reviewer #2: No

---

## [Editor Report · Acceptance letter]

30 Jun 2022

PONE-D-21-34583R1 

Direct addition of poly-lysine or poly-ethylenimine to the medium: a simple alternative to plate pre-coating 

Dear Dr. Faussner:

I'm pleased to inform you that your manuscript has been deemed suitable for publication in PLOS ONE. Congratulations! Your manuscript is now with our production department. 

Kind regards, 

on behalf of

Dr. Nazmul Haque 

Academic Editor

PLOS ONE